# Association of colitis with gut-microbiota dysbiosis in clathrin adapter AP-1B knockout mice

**Aditi Jangid[1], Shinji Fukuda[2,3,4¤a], Masahide Seki[5], Terumi Horiuchi[5], Yutaka Suzuki[5], Todd D. Taylor[6], Hiroshi Ohno[3¤b], Tulika Prakash[6¤c]***

**1** BioX Centre and School of Basic Sciences, Indian Institute of Technology Mandi, Mandi, Himachal Pradesh, India, **2** Laboratory for Intestinal Ecosystem, RIKEN Center for Integrative Medical Sciences, Yokohama, Kanagawa, Japan, **3** Intestinal Microbiota Project, Kanagawa Institute of Industrial Science and Technology, Kawasaki, Kanagawa, Japan, **4** Transborder Medical Research Center, University of Tsukuba, Tsukuba, Ibaraki, Japan, **5** Department of Computational Biology and Medical Sciences, The University of Tokyo, Kashiwa, Chiba, Japan, **6** Laboratory for Microbiome Sciences, RIKEN Center for Integrative Medical Sciences, Yokohama, Kanagawa, Japan

¤a Current address: Institute for Advanced Biosciences, Keio University, Tsuruoka, Yamagata, Japan
¤b Current address: Laboratory for Intestinal Ecosystem, RIKEN Center for Integrative Medical Sciences, Yokohama, Kanagawa, Japan
¤c Current address: BioX Centre and School of Basic Sciences, Indian Institute of Technology Mandi, Mandi, Himachal Pradesh, India
* tulika@iitmandi.ac.in

**Data Availability Statement:** All files are available from the NCBI SRA database (Bioproject ID PRJNA541820.), https://www.ncbi.nlm.nih.gov/bioproject/?term=PRJNA541820

## Abstract

Inflammatory bowel disease results from alterations in the immune system and intestinal microbiota. The role of intestinal epithelial cells (IECs) in maintaining gut homeostasis is well known and its perturbation often causes gastrointestinal disorders including IBD. The epithelial specific adaptor protein (AP)-1B is involved in the establishment of the polarity of IECs. Deficiency of the AP-1B µ subunit (Ap1m2-/-) leads to the development of chronic colitis in mice. However, how this deficiency affects the gut microbes and its potential functions remains elusive. To gain insights into the gut microbiome of Ap1m2-/- mice having the colitis phenotype, we undertook shotgun metagenomic sequencing analysis of knockout mice. We found important links to the microbial features involved in altering various physiological pathways, including carbohydrate metabolism, nutrient transportation, oxidative stress, and bacterial pathogenesis (cell motility). In addition, an increased abundance of sulfur-reducing and lactate-producing bacteria has been observed which may aggravate the colitis condition.

## Introduction

Inflammatory bowel disease (IBD) results from abnormal cross-talk between the host and intestinal microbiota in an immunocompromised or genetically susceptible individual. IBD is considered as an autoimmune disease in which the body's own immune system attacks the elements of the digestive system. IBD is characterized by chronic intestinal inflammation that can

**Funding:** This study was supported in part by Japan Society For The Promotion of Science (JSPS) KAKENHI (grant no. 18H04805 to S.F.), Japan Science and Technology Agency (JST) PRESTO (grant no. JPMJPR1537 to S.F.), JST ERATO (grant no. JPMJER1902 to S.F.), Advanced Research & Development Programs for Medical Innovation (AMED-CREST; grant no. JP19gm1010009 to S.F.), the Takeda Science Foundation (to S.F.), the Food Science Institute Foundation (to S.F.), the Program for the Advancement of Research in Core Projects under Keio University's Longevity Initiative (to S.F.). This study was also supported in part by JSPS KAKENHI (grant no. 16H06279 to Y.S.). The Council of Scientific and Industrial Research, India provided support in the form of a fellowship to AJ. TP received support in the form of salary from IIT Mandi. In addition, the HPC facility at IIT Mandi has been used for bioinformatics analysis. RIKEN provided material support and support in the form of salaries for TT and HO. The specific roles of these authors are articulated in the 'author contributions' section.

**Competing interests:** The authors have declared that no competing interests exist.

occur in both the small and large intestines. This inflammatory condition includes two main forms, ulcerative colitis (UC) and Crohn's disease (CD). In patients with IBD, high-throughput studies of intestinal microbiota using shotgun metagenomics or 16S rRNA sequencing have shown a state of dysbiosis, which is characterized by altered microbial diversity, viz., reduced phylum *Firmicutes* and enhanced *Proteobacteria* [1]. In addition, reports based on amplicon metagenomic sequencing have often exhibited imputed functional alterations and found altered metabolic pathways associated with increased nutrient transport systems, oxidative stress, and decreased amino acid metabolism [2]. However, heterogeneity in both IBD aetiology and pathology creates a complication in the generalization of microbial associations across the disease spectrum.

In order to investigate the microbial associations involved in intestinal inflammation, mouse models of IBD are the most preferred potential systems as environmental influences and host genetics can be easily controlled. Different mouse models, however, reproduce different features of human diseases and thus result in differential host-microbial interactions. Therefore, it is imperative to perform comprehensive analyses of the intestinal microbial diversity across different IBD models which will highlight various aspects of the immunological modulation of the gut microbiota and vice-versa. Towards this, community profiling studies have been carried out in several IBD models and diversity in taxonomy has been observed in the different models under investigation [3]. For example, the colitis model generated through the dextran sodium sulphate (DSS) feed leads to a chemical disruption of the mucosal epithelium, resulting in inflammation induction driven by microbiota [4]. In this model, multiple bacterial taxa have been found to increase, including *Enterobacteriaceae* [5, 6], and flagellin transcripts have been found to decrease at the functional level [6, 7]. In contrast, mouse models generated by knocking out T-bet (T-bet$^{-/-}$), a transcription factor, along with the deficiency of an acquired immune system (T-bet$^{-/-}$ Rag2$^{-/-}$ UC (TRUC)), has led to the development of spontaneous colitis, which is a consequence of increased TNF-α (dendritic cell-derived) and dysbiosis in commensal microbiota [8, 9]. The authors reported a high abundance of colitogenic microbiota in this model, including high abundance of the family *Enterobacteriaceae*, which has been linked to tetrathionate metabolism [10]. This was suggested as the consequence of the abnormal innate immune control of the commensal microorganisms [10]. In contrast to the DSS model, the TRUC model was found to be linked to an increase in flagellar components [10], suggesting heterogeneous pathogenic working mechanisms in these IBD models.

In addition to these microbiome-associated colitis studies, there are cases where the microbiome association with colitis remains unexplored at the genetic level. One such example is of the development of chronic colitis in adaptor protein (AP)-1B deficient mice. The role of intestinal epithelial cells (IECs) in maintaining intestinal homeostasis is well known. IECs lining the mucosal surface are a physical barrier between the internal environment of the host and the external environment. The highly polarized nature of the epithelial cells make them distinct, and a critical factor for maintaining this polarity is AP-1B, the epithelium-specific basolateral targeting factor [11, 12]. The function of AP-1B–mediated protein targeting in maintaining homeostasis of the gastrointestinal immune system has been previously elucidated [13]. This was done by using mice in which the gene *Ap1m2*, which is responsible for expression of the epithelial specific medium subunit (μ1B) of AP-1B protein had been knocked out. The absence of this protein spontaneously led to the development of T-helper (Th) 17-dominant colitis due to impaired functioning of the two vital epithelial effector functions: luminal transport of secretory IgA and expression of antimicrobial proteins [13]. These defects have been shown to increase bacterial translocation into the colonic mucosa [13]. The polymeric Ig receptor (pIgR) and a substantial number of basolateral cytokine receptors mistargeted to the apical plasma

membrane resulted in compromised immune function of the epithelial cells in the $Ap1m2^{-/-}$ mice. The authors also found reduced expression of the $AP1M2$ gene in the colonic epithelium of CD patients [13]. Additionally, mislocalization of the basolateral cytokine receptors was reported in one of the patient samples. All of the above phenotypes confirm that the absence of AP-1B lead to overall perturbation of gastrointestinal immune homeostasis.

In our previous study the molecular mechanisms of AP-1B in immune homeostasis were explored. However, alternations in the microbial population due to the deficiency of this protein complex has not been addressed. In addition, the effect of the absence of the $Ap1m2$ gene on the gut microbes and their potential functions in the development or progression of spontaneous chronic colitis remains unknown. Towards this, we have performed a whole genome shotgun (WGS) based metagenomic analysis of mice lacking the $Ap1m2$ gene, which is responsible for expression of the AP-1B medium subunit (μ1B) [14].

## Material and methods

### Animal experiments and sample collection

All animal experiments were approved by the Institutional Animal Care and Use Committee (IACUC) of the RIKEN Yokohama Branch. Mice were maintained under specific pathogen-free (SPF) conditions in the animal facility at the RIKEN Yokohama Branch. A total of eight fresh fecal samples were collected from four control ($Ap1m2^{+/-}$) and four knockout ($Ap1m2^{-/-}$) 10-week-old male (C57BL/6) mice. The fecal samples were stored at -80˚C before DNA extraction.

### DNA extraction

Fecal DNA extraction was performed as described previously [15]. Briefly, 10 mg of freeze-dried fecal samples were disrupted with 3 mm and 0.1 mm zirconia/silica beads by vigorous shaking (1,500 r.p.m. for 5min) using a Shake Master (Biomedical Science) suspended in DNA extraction buffer containing 200 $\mu$L of 10% (w/v) SDS/TE (10 mM Tris-HCl, 1 mM EDTA, pH8.0) solution, 400 $\mu$L of phenol/chloroform/isoamyl alcohol (25:24:1), and 200 $\mu$L of 3 M sodium acetate. After centrifugation, bacterial genomic DNA was purified by the standard phenol/chloroform/isoamyl alcohol protocol. RNAs were removed from the sample by RNase A treatment.

### Shotgun metagenomic sequencing

The sequence libraries of WGS were constructed using the Illumina TruSeq DNA Sample Preparation kit having catalog number PE-940-2001. Sequencing was performed using the Illumina HiSeq2000 platform to generate 125 bp paired-end reads. In a step of End repair, the fragments were purified using AMPureXP beads with gel-free method. Each sample was run in a single lane. This approach generated around 1 Tb of sequence data for all eight samples. The quality of raw reads was analyzed using FastQC (version- 0.11.5) [16]. Adapter removal and Q-score (Q = 30) based filtering was done using FaQCs (version- 1.34) [17]. Duplicated sequences were removed by PRINSEQ (PRINSEQ-lite 0.20.4) [18] and host DNA was removed using Bowtie2 (version 2.2.5) [19].

### Metagenomic data analysis

Data analysis of metagenomic quality filtered reads was performed using the MEGAN Community Edition (version 6.8.18) [20]. Alignment (BLASTX) of the quality filtered reads was performed using DIAMOND (version 0.9.9.110) [21] at default parameters using nr-db

(downloaded on 19 July 2017) as the reference database. The taxonomy and functional classification profiles were generated by MEGAN using a minimum score of 50 and a cutoff of the top 10%. The taxonomical and functional profiles were generated using the mapping files (prot_acc2tax-May2017.abin and acc2eggnog-Oct2016X.abin, respectively) provided on the MEGAN website. The backtracing of statistically significant functions with statistically significant taxa was done using EggNOG 4.5.1 [22]. The statistically significant differences between the control and knockout metagenomic samples were identified using STAMP (version 2.1.3) [23]. The differences between these two groups, or datasets, were analyzed using Welch's t-test. The confidence interval and p-value threshold for the analysis was set to 95% and 0.05, respectively.

Additionally, the artifacts introduced during library preparation may introduce a bias in the resulting sequences. Apart from this, the abundance of taxa neccessary to exert a physiological effect also needs to be considered. With this in mind, we set abundance percent thresholds on all taxonomy levels, which have been found to be differing significantly between the control (CO) and knockout (KO) mice, to identify those taxa with an abundance equal to or greater than 0.005% in atleast one of the samples among all eight samples. In addition to this, a threshold was set at the functional level such that only those significant functions were considered for the downstream analysis that have a mean abundance difference between CO & KO groups > 5.

## Results

We sequenced four control (CO) and four knockout (KO) fecal samples using Illumina paired-end sequencing technology. This resulted in 2,687,761,416 raw sequences with an average number of 335,970,177 reads per sample and an average read-length of 125 bps (S1 Table). After performing a quality filtering analysis, including adaptor trimming and removal of low quality reads, duplicated reads and reads matching to the mouse reference genome, 1,282,643,136 high quality reads (Q>30) with an average number of 160,330,392 reads and an average read length of 109 bps per sample were obtained.

### Identification of microbial taxa in the control and knockout groups

Taxonomic composition analysis indicated that *Firmicutes* and *Bacteroidetes* were the predominant phyla followed by *Verrucomicrobia*, *Actinobacteria*, and *Proteobacteria* in all the samples of the control (CO) and knockout (KO) groups (S1 File). However, the proportion of these taxa was found to be different in the two groups. The KO group compared to the CO group exhibited increased levels of the phyla *Firmicutes* (CO: 66.62% and KO: 79.17%), *Proteobacteria* (CO: 0.31% and KO: 0.68%), and *Verrucomicrobia* (CO: 0.023% and KO: 0.22%). The levels of the phyla *Bacteroidetes* (CO: 31.88% and KO: 18.91%) and *Actinobacteria* (CO: 0.58% and KO: 0.33%) were found to be decreased in the KO as compared to the CO group. This observation indicated that while the proportion of each bacterial group was different in the two groups, the community composition remained unchanged.

Statistical analysis on the relative abundance of the bacterial taxa identified in the two groups revealed significant differences between the gut microbiomes of the CO and KO groups. At the phylum level, all the significantly different taxa between the CO and KO groups were found to be *Candidatus* (Fig A in S2 File). Interestingly, the relative abundance of the phylum *Firmicutes* was found to be higher in the KO (S1 File) as compared to the CO group, however no statistical significance was observed. At the order level, a total of 11 taxa were found to be significantly altered between the CO and KO groups (Fig C in S2 File). Interestingly, at this level no taxa related to the phylum *Firmicutes* was found to be significantly altered

between the two groups. A total of 14 taxa were found to be significantly altered between the CO and KO groups at the family level (Fig D in S2 File). Out of these 14 taxa, two families, including *Peptostreptococcaceae* (p = 0.012) and *Sporanaerobacter* (p = 0.046), belonging to the phylum *Firmicutes*, were found to be significantly increased in the KO as compared to the CO group. The increased abundance of *Peptostreptococcaceae* has been previously reported in IBD patients [24]. At the genus level a total of 77 genera were found to be significantly altered between the two groups (Fig E in S2 File). Out of these 77 genera, 11 taxa belonging to the phylum *Firmicutes*, including *Romboutsia* (p = 0.016), *Tepidimicrobium* (p = 0.004), *Ezakiella* (p = 0.009), *Terrisporobacter* (p = 0.022), *Tuberibacillus* (p = 0.023), *Hydrogenibacillus* (p = 0.03), *unclassified Peptostreptococcaceae* (p = 0.038), *Facklamia* (p = 0.039), *Nosocomiicoccus* (p = 0.039), *Geomicrobium* (p = 0.042), and *Vulcanibacillus* (p = 0.044), were found to be increased in the KO as compared to the CO group. At the species level, a total of 551 taxa, including 164 species of the phylum *Firmicutes*, were found to be significantly altered between the CO and KO groups (S2 Table). A large majority of the species belonging to the phylum *Firmicutes* (138, 25.04%) were found to be increased in the KO as compared to the CO group (26, 4.71%) (S2 Table). The members belonging to the *Lactobacillus* species were found to be significantly increased in the KO group (KO: 25 and CO: 2 out of 164). Similarly, the number of species related to the order *Clostridiales* was significantly increased in the KO group (KO: 38 and CO: 3 out of 164). This observation corroborates with the previous study where an increased abundance of *Clostridiales* has been reported in case of colitis [25].

An alteration in the relative abundance of the taxa belonging to the phylum *Actinobacteria*, *Proteobacteria* and *Verrucomicrobia* were also observed in our analysis (Fig C-E in S2 File). In the phylum *Actinobacteria*, 54 species of this phylum were found to be altered between the two groups (S2 Table). In a previous study, the phylum *Actinobacteria* was shown to be increased in the remission phase of colitis patients [10]. In the phylum *Proteobacteria*, at the species level, a total of 179 species were found to be altered between the two groups (S2 Table). It is interesting to note that the species related to the class *Deltaproteobacteria* were significantly increased in the KO group, while the species belonging to the class *Gammaproteobacteria* were decreased in the KO group as compared to the CO group. The class *Deltaproteobacteria* contains bacteria that reduce sulfate. The higher abundance of these bacteria is also known in IBD patients. Sulfate-reducing bacteria have been suggested to aggravate gastrointestinal diseases by making hydrogen sulfide ($H_2S$) and other harmful by-products as well as by reducing beneficial metabolites, such as butyrate [26]. Apart from this, species belonging to the classes *Alpha-*, *Beta-*, and *Epsilon- proteobacteria* were found to be significantly changed between the two groups. The species *Helicobacter saguini* belonging to the class *Epsilonproteobacteria* was isolated from the intestines and feces of the new world monkey "cotton-top tamarins (CTTs)" with chronic colitis [27].

The phylum *Bacteroidetes* was not found to be significantly different at any of the taxonomic levels between the CO and KO groups, except for at the species level (S2 Table). In total 551 species were found to significantly differ between the CO and KO mice. Out of these 551, the relative abundances of 218 species were found to be increased in the CO group, whereas that of 333 species were found to be increased in the KO group (S3 File and S2 Table). Upon investigating the taxonomic hierarchy of the significantly altered species, we observed that the numbers of species belonging to the phylum *Firmicutes* and *Bacteroidetes* were similar in the CO group. However, in the KO group the species belonging to the phylum *Firmicutes* were found to have profoundly increased, whereas the species belonging to the phylum *Bacteroidetes* were found to have decreased as compared to the CO mice. This observation corroborates with a previous study in which a decreased abundance of the species belonging to the phylum *Bacteroidetes* is reported in the case of chemically induced colitis [25].

Sequencing related artifacts are known to greatly impact the abundance of taxa in metagenomic sequencing [28]. To avoid this bias in our results, we discarded the less abundant reads (relative abundance < 0.005%) and re-assessed the significantly altered taxa between the CO and KO groups at all taxonomic levels (Table 1). We further explored the roles of these microbial species with respect to the formation of microbial fermentation products (butyrate, propionate, acetate, lactate and $H_2$ pathways) using a comprehensive literature survey and gene content analysis. We explored these SCFAs mainly because of their potential involvement in gut homeostasis. The complete set of genes associated with butyrate and propionate production were not found in any of the mentioned species (S3 Table). The genes associated with lactate and acetate production were found in some species (~71%, 25%, respectively) while absent in others (~29%, 75%, respectively) (S3 Table). Additionally, we also correlated the involvement of these micro-organisms in previous colitis studies (Table 1).

## Identification of microbial functions in the control and knockout groups

The gut microbiota plays important roles in the physiological functions associated with nutrition, the immune system, and defense mechanisms of the host. However, dysbiosis in the microbial population caused by gene knockout in the host might lead to altered functions of the gut microbiota. To investigate the differences, if any, in the functional compositions between the CO and KO groups, we performed a comprehensive functional analysis of the WGS metagenomic reads. The functional annotations were obtained at three hierarchical levels. At level 1, the functions were grouped into three major categories including metabolism, cellular processes and signaling, and information storage and processing. A majority of the reads were mapped to the genes belonging to the metabolism category (Fig A in S4 File), however, no significant difference was observed between the CO and KO groups.

At levels 2 and 3 significant differences were observed between the CO and KO groups. Level 2 encompasses all the pathways corresponding to level 1 categories. At level 2, one functional category, viz., "carbohydrate transport & metabolism", was found to be decreased in the KO group as compared to the CO group (Fig B and C in S4 File). Level 3 encompasses all COGs/NOGs that come under the pathways in level 2. To explore the differences in the CO and KO groups based on their functional compositions, a Principal Component Analysis (PCA) was performed. At levels 1 and 2 not much clear separation was obtained in the PCA analysis (Fig 1), however, at level 3 the CO and KO groups were found to be separated between PC1 and PC3 axes (which explains 76.3% of the variation in the data). These observations suggests that the resident gut microbiota of the two groups might perform their functions differentially. In total 245 functions at level 3 were found to be significantly altered (S4 Table). Out of these functions, 44 functions were found to have very low read counts (mean abundance difference between CO & KO groups < 5) and were not considered for further analysis. The details of the results obtained for the remaining 201 significantly altered functions are given in S5 Table. Out of these 201 COGs/NOGs, 128 were found to be increased while 73 were found to be decreased in the KO as compared to the CO group. A comprehensive literature survey was performed on these 201 functions to see the effect of these functional alterations in IBD. Table 2 lists a few selected microbial functions which were found to be significantly altered in the CO and KO groups. For example, the genes involved in carbohydrate metabolism, including alpha-galactosidase (COG3345), were found to be significantly reduced in the KO group. This gene participates in galactose metabolism, glycosphingolipid biosynthesis-globo series, sphingolipid metabolism and glycerolipid metabolism. Similarly, another gene, aldolase (COG0191), which is known to be involved in several metabolic processes including fructose and mannose metabolism, pentose phosphate pathway, and biosynthesis of amino acids, was

**Table 1. List of significantly altered taxa with a relative abundance of at least 0.005% between the CO and KO groups at all taxonomic levels.** Statistical significance was tested using Welch's t-test with the confidence interval threshold of 95% and p<0.05.

| | Taxa | p-value |
|---|---|---|
| *Firmicutes* > *Clostridia* > *Clostridiales* [a] [25] | | |
| Family | *Peptostreptococcaceae* [c] | 0.012 |
| Genus | *Romboutsia* [c] | 0.016 |
| | *Terrisporobacter* [c] | 0.022 |
| | *Unclassified peptostreptococcaceae* [c] | 0.038 |
| Species | *Romboustsia ilealis* [c] | 0.021 |
| | *Romboutsia timonensis* [c] | 0.039 |
| | *Terrisporobacter glycolicus* [c] | 0.008 |
| | *[Clostridium] dakarense* [c] | 0.032 |
| | *[Clostridium] hiranonis* [c] | 0.008 |
| | *Eubacterium sp. 3_1_31* [c] | 0.022 |
| | *Eubacterium sp. CAG:841* [c] | 0.012 |
| | *Clostridium sp. CAG:221* [c] | 0.017 |
| *Firmicutes* > *Bacilli* > *Lactobacillales* > *Lactobacillaceae* > *Lactobacillus* | | |
| Species | *Lactobacillus intestinalis* [c] | 0.021 |
| | *Lactobacillus sp. ASF360* [c] | 0.02 |
| | *Lactobacillus iners* [c] | 0.038 |
| | *Lactobacillus amylovorus* [c] | 0.042 |
| | *Lactobacillus acidophilus* [c] | 0.034 |
| *Proteobacteria* > *Deltaproteobacteria* > *Desulfovibrionales* > *Desulfovibrionaceae* [a] [29] | | |
| Genus | *Mailhella* [c] | 0.050 |
| Species | *Desulfovibrio sp. 3_1_syn3* [c] | 0.021 |
| | *Desulfovibrio fairfieldensis* [c] | 0.043 |
| *Firmicutes; Erysipelotrichia; Erysipelotrichales; Erysipelotrichaceae; unclassified Erysipelotrichaceae; unclassified Erysipelotrichaceae (miscellaneous)* [b] [30] | | |
| Species | *Erysipelotrichaceae bacterium 5_2_54FAA* [c] | 0.007 |
| *Bacteroidetes; Bacteroidia; Bacteroidales; Odoribacteraceae; Odoribacter* | | |
| Species | *Odoribacter splanchnicus CAG:14* [c] | 0.037 |
| **Environmental Samples** | | |
| Species | *Firmicutes bacterium CAG:102* [c] | 0.041 |
| | *Firmicutes bacterium CAG:145* [d] | 0.036 |
| | *Firmicutes bacterium CAG:41* [c] | 0.026 |
| *Firmicutes; Bacilli; Lactobacillales; Streptococcaceae; Streptococcus* | | |
| Species | *Streptococcus suis* [d] | 0.043 |
| | *Streptococcus equinus* [d] | 0.020 |
| *Bacteroidetes; Flavobacteriia; Flavobacteriales; Flavobacteriaceae; Capnocytophaga* | | |
| Species | *Capnocytophaga sp. oral taxon 332* [d] | 0.034 |
| *Bacteroidetes; Bacteroidia; Bacteroidales; Prevotellaceae; Prevotella* | | |
| Species | *Prevotella stercorea* [d] | 0.046 |

[a] The taxonomic rank has been shown to be involved in colitis in previous studies

[b] The taxonomic rank has been shown to be involved in colitis in previous studies, however, their role remains unclear

[c] Indicates the abundance of a particular taxa is increasing in KO as compared to CO.

[d] Indicates the abundance of a particular taxa is decreasing in KO as compared to CO.

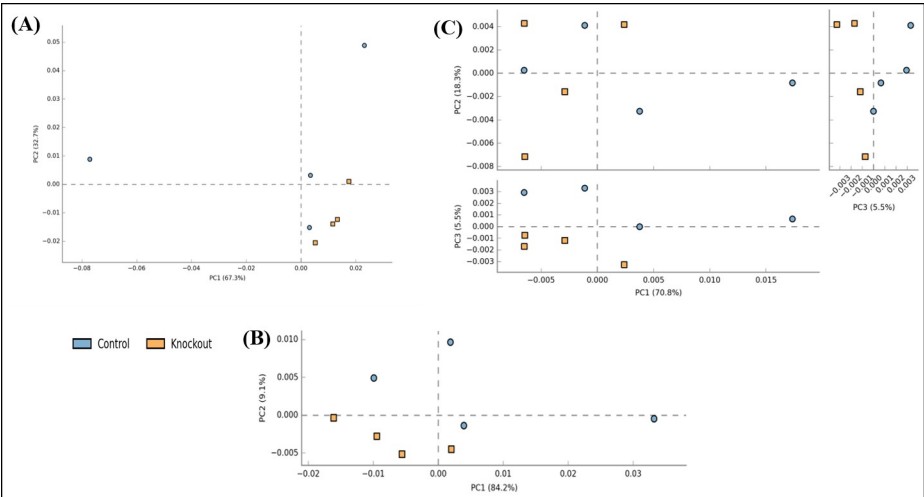

**Fig 1.** Principal component analysis (PCA) plot of the functional profiles of CO and KO groups at (A) level 1, (B) level 2, and (C) level 3 of EggNOG. Each dot represents a sample and the distinct color represents the type of the sample. The percent variability explained by each principal component is shown in parentheses in the axes legends.

found to be decreased in the KO group. Yet another set of genes, which are involved in the process of initiation and elongation of fatty acid biosynthesis (COG0332, COG0615) were found to be decreased in the KO group. Additionally, genes related with amino acid biosynthesis viz, proline (COG0014) and histidine (COG0040, COG1387) were often found to be decreased in the KO as compared to the CO group.

Certain functions were found to be increased in the KO as compared to the CO groups. For example, the gene belonging to the ycfa family (COG1724), which is involved in cell motility, was found to be increased in the KO group. The enzyme endo-beta-n-acetylglucosaminidase

**Table 2. List of selected significantly altered functions in the CO and KO groups which might be involved directly and indirectly in the progression of colitis.** The confidence interval threshold was 95% and p<0.05. Dark grey color indicates the abundance of a particular function is increasing and light grey color indicates the abundance of a particular function is decreasing.

| Genes involved in functions | CO | KO |
|---|---|---|
| Carbohydrate metabolism [31] [f] | | |
| Lipid metabolism [f] | | |
| Cellwall/membrane/envelope biogenesis [f] | | |
| Cell motility [10, 31] [e] | | |
| Mucin degradation [32] [e] | | |
| Replication, recombination and repair [e] | | |
| Nucleotide transport and metabolism [e] | | |
| Transposase [e] | | |
| Amino acid transportation [2] [e] | | |
| Carbohydrate transportation [e] | | |
| Iron transportation [33] [e] | | |
| Oxidative stress [34, 35] [e] | | |
| β-lactamase [36] [e] | | |
| Proline Biosynthesis [37] | | |
| Histidine Biosynthesis [37] | | |
| Arginine degradation [37] | | |

(COG4724, COG4193) which is known for its enzymatic activity in mucin degradation, was also found to be increased in the KO group. Apart from these, some other genes, for example, the enzymes involved in DNA primase activity (COG0358), precursors necessary for DNA synthesis (COG1372), enzymes involved in DNA repair mechanism (COG4294, COG0322), etc. were found to be increased in the KO group. Furthermore, ABC transporters (COG0411, ENOG4111GBW), major facilitators (ENOG41105XQ, ENOG410XNZG, ENOG410XSHZ, ENOG410ZWA1), tonB dependent receptors (ENOG410ZNX6, ENOG410ZPJF, ENOG410ZJX3, ENOG410YNPF, ENOG410YAIR, ENOG410YB7D), tonB dependent receptor plugs (ENOG410XQAZ, ENOG410YB3G, ENOG4110QZN), etc. were found to be increased in the KO group. The genes involved in oxidative stress control, glutathione metabolism (COG0260), transcriptional regulator, crp fnr family (ENOG410ZDRF), and osmC family (ENOG4111UEA) were also found to be increased in the KO group. Additionally, genes involved in degradation of arginine (COG0010, COG2235) and histidine (COG1228) were also found to be increased in the KO group as compared to the CO group.

### Backtracing from the microbial functions to the microbial taxa

Alterations in the functional composition are expected to be a result of the alterations in the underlying microbial populations. Thus, we wanted to explore which significant functional changes might be exerted by which significantly differing bacterial taxa in the CO and KO groups. Towards this, we backtraced the significantly differing COGs/NOGs functional classes to their corresponding bacterial taxa. Out of the 201 significantly differing COGs/NOGs functions, 89 mapped to five bacterial species, which were also found to be significantly different between the CO and KO groups. These species included, *Clostridium sp*. *BNL1100*, *Paenibacillus sp*. *Y412MC10*, *Erysipelotrichaceae bacterium 5_2_54FAA*, *Desulfovibrio sp*. *3_1_syn3*, and *Nitrobacter sp*. *Nb-311A* (S6 Table). Out of these five species, two species, *Erysipelotrichaceae bacterium 5_2_54FAA* and *Desulfovibrio sp*. *3_1_syn3*, were found to have a relative abundance higher than 0.005%.

It has been reported previously [29] that an excessive hydrogen sulfide ($H_2S$) production is found in the feces of ulcerative colitis patients. In this way, we explored the genes associated with $H_2S$ production pathways (using BioCyc and Microscope) in the above-mentioned bacteria which may eventually lead to colitis progression. The investigated pathways were sulfate reduction I (assimilatory), sulfate reduction IV (dissimilatory), sulfate reduction V (dissimilatory) and sulfur reduction II (via polysulfide) (S7 and S8 Tables). The two pathways for $H_2S$ production, viz., sulfur reduction II (via polysulfide) and sulfate reduction I (assimilatory) has been observed to be completely present in *Desulfovibrio sp*. *3_1_syn3* and *Nitrobacter sp*. *Nb-311A*, respectively (Table 3). Moreover, a 75% complete sulfate reduction I (assimilatory) pathway was found in *Clostridium sp*. *BNL1100*, *Desulfovibrio sp*. *3_1_syn3*, and *Paenibacillus sp*. *Y412MC10* (S7 and S8 Tables). Apart from this, a 60% complete sulfate reduction IV (dissimilatory) pathway was found in *Erysipelotrichaceae bacterium 5_2_54FAA* (S7 Table).

## Discussion

In this study, we demonstrated that the *Ap1m2 $^{-/-}$* (KO) mice have a different gut-microbiome pattern as compared to the control (CO) mice. We applied the WGS metagenomic method to discover the gut microbiome alterations in the two groups of mice (CO and KO) and identified the potential gut microbiome biomarkers, including composition and function of the microbiome, in the KO mice group. Our previous study indicated that the KO mice developed spontaneous chronic colitis, which was characterized by the acquisition of IL-17A and TNF-α

**Table 3. Details of genes associated with sulfur reduction II (via polysulfide) and sulfate reduction I (assimilatory) pathways found in *Desulfovibrio sp. 3_1_syn3* and *Nitrobacter sp. Nb-311A*, respectively.**

| Bacteria | EC Number | Genes |
|---|---|---|
| *Desulfovibrio sp. 3_1_syn3* | 1.97.1.3 (sulfur reductase) | ADDR02_v2_10173-sudA |
| | | ADDR02_v2_10172-sudB |
| *Nitrobacter sp. Nb-311A* | 1.8.4.8 (phosphoadenylyl-sulfate reductase) | NIT1Av1_30234-cysH |
| | 2.7.1.25 (adenylyl-sulfate kinase) | NIT1Av1_30237 |
| | 2.7.7.4 (sulfate adenylyltransferase) | NIT1Av1_30236-cysD |
| | 1.8.1.2 (sulfite reductase [NADPH]) | NIT1Av1_10139-cysJ |
| | | NIT1Av1_10138-cysI |

production in these animals [13]. All together, these outcomes stipulate that the gut-microbiome structure might contribute in disease development in mice lacking the *Ap1m2* gene.

We found that the lactate-producing bacteria (LPB) and the sulfate-reducing bacteria (SRB) were enriched in the KO group. The pro-inflammatory nature of the SRB bacteria has been reported in a number of immune or inflammatory diseases, including metabolic syndrome [38], type 2 diabetes (T2D) [39], and IBD [40]. The SRB bacterial species impede butyrate β-oxidation and deteriorate butyrate [40, 41]. Butyrate is a critical metabolite among the SCFAs as it protects the intestinal epithelial barrier integrity and preserves the immune homeostasis of the host by inducing regulatory T cell differentiation [42, 43]. A decreased level of butyrate causes dysfunction in the intestinal epithelium barrier and leads to the expression of several inflammatory components such as the pathogen-associated molecular pattern (PAMP) or the microbe-associated molecular pattern (MAMP) [44]. Both PAMP and MAMP can affect the IECs [44]. In addition, $H_2S$, a cytotoxic byproduct of SRB that exerts pro-inflammatory effects at high concentrations [45, 46], can exacerbate intestinal epithelial barrier damage and its excessive production is reported in colitis condition [26]. High $H_2S$ levels and low butyrate levels were detected when butyrate producing bacteria (BPB) and SRB were co-cultured with lactate-producing bacteria [47]. Based on these observations, we hypothesize that because SRB can utilize hydrogen and lactate as the substrates for $H_2S$ production, SRB may compete for these substrates with the hydrogen consuming bacteria and lactate-utilizing bacteria such as methanogens and BPB. A similar hypothesis has been proposed in the case of Behcet's disease, which is a multisystemic inflammatory disease that causes inflammation in blood vessels (throughout the body) including the digestive system [48].

In a previous study it was observed that in $Ap1m2^{-/-}$ mice there is a decrease in the numbers of goblet cells which resulted in reduced mucus secretion [13]. In this study, we aimed to explore the effect of this reduced mucus secretion in terms of microbial shift in the gut. One of the most important structural components of mucus is the mucin glycoprotein. These mucins play a multifaceted and integral role in the communication between epithelial surfaces and microbes [49]. The mucin glycoproteins provide protection to the colonic epithelium and are either secreted from the goblet cells into the intestinal lumen (e.g., MUC2 and MUC5B) or are membrane-attached (e.g., MUC1 and MUC4). The secreted mucins are also involved in the formation of a two layer protective boundary in the colon. Among the two layers, one is a tightly-adherent mucus layer where bacterial exposure is strictly restricted, whereas the other is only a loosely-adherent mucus layer where intestinal bacteria find a niche. Indeed, several gut resident bacteria possess mucin-degrading abilities [50]. Towards this, we have observed an increased abundance of the mucin degrading enzyme endo-beta-n-acetylglucosaminidase, in the KO group. It has been previously observed that polysaccharide-utilization loci (PULs) involving genes encoding for the putative glycoside hydrolases (GHs) such as endo-β-N-

acetylglucosaminidase, α-L-fucosidase, α-mannosidase, and endo-β-galactosidase, have been up-regulated in *B. thetaiotaomicron* when it was grown in monoxenic mice or on mucin *O*-glycans as compared to the *in vitro* glucose control [32]. Notably, in our case, the abundance of *B. thetaiotaomicron* was reduced in the KO group thus suggesting the involvement of other microbial members in this activity. Upon backtracing the taxa responsible for contributing the mucin degrading function, we found *Paenibacillus sp.* Y412MC10 and *Erysipelotrichaceae bacterium 5_2_54FAA* to be associated with endo-β-N-acetylglucosaminidase. These species were found to be increased in the KO group and thus might act in a similar manner to *B. thetaiotaomicron* and result in further reduction of the mucin glycoproteins which might eventually aggravate colitis.

The alterations in the abundance of certain genes in the KO group suggest the role of microbiota to retain homeostasis during inflammation. For example, the abundance of oxidative stress related genes is found to increase, whereas that of carbohydrate metabolism related genes is found to decrease in the KO group mice. Previous studies have also shown an increase in the amino acid transporter genes [2] and a decrease in carbohydrate metabolism [31] genes in IBD. This suggests that the bacteria under an inflammatory state usually have a weak capacity to make nutrients on their own. To overcome the problem of nutrient deficiency, the bacteria instead transport them from the available environment through means such as tissue destruction or from the sites of inflammation. Our results are consistent with this hypothesis as the genes related to amino acid transport and carbohydrate transport are found to be increased in the KO group.

For the growth and maintenance of barrier function and mucosal integrity, amino acids are essentially required. Amino acid metabolism in microbial community of gut, plays significant role in physiology and nutrition of the host. We found genes related with proline and histidine biosynthesis decreased and arginine degradation genes increased in the KO as compared to the CO groups. In previous studies the supplementation of histidine, proline, and arginine has reported to improve IBD [37]. Dietary proline supplementation improved the production of mucin, restored healthy microbiota, protected gut epithelium, and encouraged mucosal rebuilding in DSS-treated rat [37]. Similarly, histidine supplementation improved colitis condition in an IL-10-deficient Crohn's disease model. This improvement was done by regulation of NF-κB activation, followed by inhibition of the proinflammatory cytokine secretion by macrophages [37]. Additionally, changes in the metabolism of arginine have been reported in animal colitis models as well as IBD patients, and experimental colitis has been improved with arginine supplementation [37].

In addition, the genes for glutathione metabolism were also increased in the KO group. Glutathione, which is found to be mainly synthesized by *Proteobacteria*, permits bacteria to maintain homeostasis in oxidative stress environments [35]. Towards this, we also observed an increased abundance of the genus *Mailhella* belonging to phylum *Proteobacteria* in the KO group. Moreover, upon backtracing we found that the glutathione metabolism related genes were being contributed by the microbial species including *Desulfovibrio sp. 3_1_syn3*, *Paenibacillus sp. Y412MC10*, *Nitrobacter sp. Nb-311A*, and *Erysipelotrichaceae bacterium 5_2_54FAA*. This indicates that these species may be capable of maintaining homeostasis in the inflamed gut. A few transcriptional regulators, which are known to be involved in controlling oxidative damage [34], including crp fnr family and osmC family, have also been found to be increased in the KO group. Taken together these observations suggest a significant role of the gut microbiome in maintaining overall homeostasis of the inflamed sites.

In previous IBD studies on T-bet$^{-/-}$ Rag2$^{-/-}$, resulting in colitis and TNBS-induced colitis mouse models, an increased capacity for bacterial pathogenesis has been observed. These mouse models have been found to exhibit a high abundance of genes related to cell motility

and secretion, flagellar assembly, and bacterial motility proteins [10, 31]. In our case, we also observed an increased abundance of the genes related to cell motility in the KO group. These observations suggest increased pathogenic bacterial colonization in $Ap1m2^{-/-}$, resulting in the colitis condition. It has been previously reported that anemia is one of the most frequent complications or extraintestinal manifestation of IBD. Iron deficiency is the most important cause of anemia in ulcerative colitis patients [33]. Our results are consistent with this observation as the iron sequestration genes are found to be significantly increased in the KO group. Thus, we hypothesize that enhanced iron sequestration by the resident gut bacteria leads to an anemic condition in the KO group having chronic colitis as a result of the $Ap1m2^{-/-}$ gene knockout. A similar hypothesis has been proposed earlier where the levels of heme transporter genes were observed to be elevated in DNR mice as IBD develops [51].

Our results have also demonstrated significant changes in various other pathways and gene abundances between the CO and KO groups. For example, the genes associated with "lipid metabolism" and "cell wall/membrane/envelope biogenesis" were found to be decreased while those related to "replication, recombination and repair" and "nucleotide transport and metabolism" were found to be increased in the KO as compared to the CO group. This indicates that due to reduced lipid metabolism the lipids were not available for cell wall/membrane/envelope biogenesis which brought bacterial endangered survival. Thus, bacteria might choose alternative means for handling this situation by increasing the replication rate to increase their numbers. In addition, due to an increased number of movable elements, i.e. transposase, the bacteria might allow mutational changes in their genome to adapt to an inflamed intestinal environment. Although the genes associated with "cell wall/membrane/envelope biogenesis" were found to be decreased in the KO group but the genes associated with beta-lactamase were found to be increased, this could provide a defense to the resident gut bacteria of the KO mice. The beta-lactamase produced by bacteria is known for its multi-resistance to β-lactam antibiotics. Recently it has been shown that increased β-lactamase production by gut bacteria might aggravate ulcerative colitis in patients [36]. Thus beta-lactamase may enhance colitis in $Ap1m2^{-/-}$ mice as well, but the mechanism remains unknown.

As is evident from the above discussion that a large number of the functional alterations between the CO and KO groups correlated well with the observed processes known to play a role in IBD. It is important to note that a few functional alternations still remained inconclusive. The analysis presented in the paper provides future directions for the design of more targeted experiments for the validation of the functional alterations due to the knockout of Ap1 gene leading to spontaneous colitis in mice. This is the first prospective study using next-generation sequencing method to examine the fecal microbiota composition in $Ap1m2^{+/-}$ and $Ap1m2^{-/-}$ mice and highlights the role of the gut microbiome in the colitis phenotype induced due to the gene knockout. Our analyses has elucidated the important features of microbiome dysbiosis and dysfunction in terms of taxa and gene functions in $Ap1m2^{-/-}$ mediated colitis. It is evident from this study that for a physiological condition, it is not a single species or taxa which is responsible but a group of species or taxa are usually responsible for a diseased condition. However, it remains to be concluded whether the microbiota is the cause or an effect of this disease.

## Supporting information

**S1 File. Plot representing the taxonomic affiliations at the phylum level of the reads from the KO and CO groups.** The inset shows the relative abundance on a different scale for the phyla *Verrucomicrobia*, *Actinobacteria*, *Proteobacteria*, and others.
(PDF)

**S2 File.** Extended error bar chart representing the differences in the mean proportions between the CO (blue) and KO (orange) groups at the level of (A) phylum, (B) class, (C) order, (D) family, and (E) genus. Statistical significance was tested using the Welch's t-test with $p < 0.05$
(PDF)

**S3 File. Distribution of significant species in the CO and KO groups.** (A) Distribution of 551 species, (B) Distribution of the number of species belonging to the phyla *Firmicutes* and *Bacteroidetes* in the CO and KO Groups.
(PDF)

**S4 File.** Distribution of the number of metagenomic reads in the CO and KO groups (A) in the three functional classes at level 1, (B) in different functional classes at level 2 of EggNOG-based classification. (C) Extended error bar plot representing the differences in the mean proportions of the functional category "carbohydrate transport & metabolism" between the CO and KO groups. Statistical significance was tested using the Welch's t-test with $p < 0.05$.
(PDF)

**S1 Table. Summary of metagenomic reads before and after quality filtering.**
(XLSX)

**S2 Table. Significantly altered species between the CO and KO groups.** Blue represents the taxa which are more abundant in the CO group and yellow represent those which are more abundant in the KO group.
(XLSX)

**S3 Table. Details of the genes involved in short chain fatty acid production and $H_2$ pathway in species having 0.005% abundance threshold.** Source: NCBI Genome.
(XLSX)

**S4 Table. Significantly altered functions at level 3 between the CO and KO groups.** Blue represents the functions which are more abundant in the CO group and yellow represent those which are more abundant in the KO group.
(XLSX)

**S5 Table. Details of the roles of significantly altered functions across CO & KO groups.**
(XLSX)

**S6 Table. Summary of 89 significantly altered COGs/NOGs functions which mapped to five bacterial taxa, which were also found to be significantly different between the CO and KO groups.**
(XLSX)

**S7 Table. Details of the genes involved in the sulfate reduction pathway in back-traced significant bacteria.** Source: BioCyc.
(XLSX)

**S8 Table. Details of the genes associated with different sulfate and sulfur reducing pathways in *Desulfovibrio sp. 3_1_syn3*.** Source: Microscope.
(XLSX)

## Acknowledgments

We want to thank Mr. Gaurav Chetal, who helped us in data analysis.

## Author Contributions

**Conceptualization:** Shinji Fukuda, Todd D. Taylor, Hiroshi Ohno, Tulika Prakash.

**Data curation:** Aditi Jangid, Shinji Fukuda, Masahide Seki, Terumi Horiuchi, Yutaka Suzuki, Tulika Prakash.

**Formal analysis:** Aditi Jangid.

**Funding acquisition:** Yutaka Suzuki, Tulika Prakash.

**Methodology:** Aditi Jangid, Shinji Fukuda, Yutaka Suzuki, Hiroshi Ohno, Tulika Prakash.

**Resources:** Tulika Prakash.

**Software:** Aditi Jangid.

**Supervision:** Tulika Prakash.

**Validation:** Aditi Jangid, Todd D. Taylor, Tulika Prakash.

**Writing – original draft:** Aditi Jangid, Tulika Prakash.

**Writing – review & editing:** Aditi Jangid, Todd D. Taylor, Tulika Prakash.

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
