## [Decision Letter · Decision Letter 0]

6 Nov 2019

PONE-D-19-26965

Dysbiosis in Gut-Microbiome of Clathrin Adapter AP-1B Knockout Leading to Colitis in Mice.

PLOS ONE

Dear Dr. Prakash,

Thank you for submitting your manuscript to PLOS ONE. After careful consideration, we feel that it has merit but does not fully meet PLOS ONE’s publication criteria as it currently stands. Therefore, we invite you to submit a revised version of the manuscript that addresses the points raised during the review process.

You will see that the reviewers have suggested that you discuss further on the possible functional implications of the changes observed in your study.

We would appreciate receiving your revised manuscript by Dec 21 2019 11:59PM. To enhance the reproducibility of your results, we recommend that if applicable you deposit your laboratory protocols in protocols.io, where a protocol can be assigned its own identifier (DOI) such that it can be cited independently in the future. For instructions see: http://journals.plos.org/plosone/s/submission-guidelines#loc-laboratory-protocols

We look forward to receiving your revised manuscript.

Kind regards,

François Blachier, PhD

Academic Editor

PLOS ONE

Journal Requirements:

This research was supported/partially supported by the Indian Institute of technology Mandi, India, Council of Scientific and Industrial Research, India, and RIKEN Japan. We want to thank Mr. Gaurav Chetal, who helped us in data analysis.

Y.S.

JSPS KAKENHI (16H06279)

Japan Society For The Promotion of Science

https://www.jsps.go.jp/english/e-grants/

Yes, Contributed in Methodology Section

Reviewers' comments:

Reviewer's Responses to Questions

**Comments to the Author**

1. Is the manuscript technically sound, and do the data support the conclusions?

Reviewer #1: Yes

Reviewer #2: Yes

2. Has the statistical analysis been performed appropriately and rigorously? 

Reviewer #1: I Don't Know

Reviewer #2: Yes

3. Have the authors made all data underlying the findings in their manuscript fully available?

Reviewer #1: Yes

Reviewer #2: Yes

4. Is the manuscript presented in an intelligible fashion and written in standard English?

Reviewer #1: Yes

Reviewer #2: Yes

5. Review Comments to the Author

Reviewer #1: This study shows comprehensive analyses of the microbiome in Ap1m2 KO mice, which the authors previously demonstrated to develop chronic colitis and epithelial cell polarity defects. Loss of AP1M2 expression has also been shown in patients diagosed with IBD.

The authors made applaudabe efforts to characterization and bioinfiormatically analyze all the observed changes between control and KO mice. However, the functional significance of these changes is not experimentally addressed. With so many things changing it will be difficult to determine which of the changes are relevant and which are not. Selected agreements between observed changes in the microbiome and therewith associated processes and processes already known to play a role in IBD knowledge may be coincidental. I wonder how many of the observed changes in the microbiome and therewith associated processes did not correlate with processes known to play a role in IBD. Further, some discussion on the etiology of the microbial changes would be welcome. How specific are these for Ap1m2 KO mice, when compared to other mouse models of colitis? Finally, the title of this manuscript ('Dysbiosis in Gut-Microbiome of Clathrin Adapter AP-1B Knockout Leading to Colitis in Mice), as well as the running title (Role of Gut-Microbiome in AP-1B Knockout Colitis), implies that dysbiosis leads to colitis in these mice, but there is no evidence presented to support such causality.

Taken together, I think this work will provide a useful microbiome reference for future studies but, as such, does not provide information about the functional implications of the observed microbiome changes.

Reviewer #2: This study uses shotgun metagenomic sequencing of the gut microbiota in control and Ap1m2-/- mice. They found a shift in some of the taxa, that may be associated with colitis progression and/or disease phenotype in the KO mice. Overall this data is a nice description of the changes in microbiota in the KO mice, and underscores the importance of AP-1B expression in intestinal epithelial cells.

6. PLOS authors have the option to publish the peer review history of their article (what does this mean?). If published, this will include your full peer review and any attached files.

Reviewer #1: No

Reviewer #2: No

---

## [Author Response · Author response to Decision Letter 0]

20 Dec 2019

Reviewer #1: This study shows comprehensive analyses of the microbiome in Ap1m2 KO mice, which the authors previously demonstrated to develop chronic colitis and epithelial cell polarity defects. Loss of AP1M2 expression has also been shown in patients diagosed with IBD.

The authors made applaudabe efforts to characterization and bioinformatically analyze all the observed changes between control and KO mice. 

1. However, the functional significance of these changes is not experimentally addressed. With so many things changing it will be difficult to determine which of the changes are relevant and which are not. Selected agreements between observed changes in the microbiome and therewith associated processes and processes already known to play a role in IBD knowledge may be coincidental. I wonder how many of the observed changes in the microbiome and therewith associated processes did not correlate with processes known to play a role in IBD. 

Reply:

We understand the concern raised by the reviewer very well. We would like to reinstate that the phenotype observed in the knockout mice is spontaneous chronic colitis [1]. The objective of this study is mainly to see the alterations in the entire microbiota [taxonomic and functional composition] due to the gene knockout. Towards this, we have used the whole metagenome shotgun sequencing approach, which is highly preferred to study the functional aspects of a system in addition to the taxonomic features. Having identified the significantly different functions between the two groups (CO and KO), we performed an in-depth literature review to further understand their roles in details. In total 245 functions at level 3 were found to be significantly altered. Out of these functions, 44 functions were found to have very low read counts (mean abundance difference between CO & KO group < 5) and were not considered for further analysis. The details of the results obtained for the remaining 201 significantly altered functions are given in Table 1 below. 

In the previous version of the manuscript, we had only discussed a few of these significantly altered functions in detail. This perhaps led to the impression that only selected observed functional alterations in the microbiome correlated with the processes already known to play a role in IBD knowledge and might be coincidental. Towards this, we have now provided the details of all the 201 significantly altered functions as S5 Table in the revised version of the manuscript. As is evident from this Table 1 below (S5 Table) that a large number of these functional alterations correlated well with the observed processes known to play a role in IBD. It is indeed true that only a few functional alternations remained inconclusive. The analysis presented in the paper provides future directions for the design of more targeted experiments for the validation of the functional alterations due to the knockout of Ap1 gene leading to spontaneous colitis in mice.

Table 1: Details of the roles of significantly altered functions across CO & KO groups.

Microbiome Functions CO KO Correlation to colitis phenotype

Oxidative stress & Nitrosative stress resistance significantly increased in KO mice group

Oxidative stress [2, 3]

 ENOG41125EY, COG0450, ENOG410YVS9,

ENOG4112704 COG0260, ENOG4111TDN, ENOG4111IWY, ENOG4111UEA, COG1233, ENOG410ZDRF, COG2344, ENOG41123KF, ENOG4111K4T, ENOG410XVW8, ENOG4111KHV, ENOG4111QAE During colitis condition mice gut is exposed to oxidative and nitrosative stress, these enzymes

regulate oxidative stress and nitrosative stress in bacteria

Nitrosative stress [4]

 COG1017, ENOG410ZDRF(repeated) 

Transporters significantly increased in KO mice group, i.e. more nutrient uptake in KO mice group

Iron Uptake/transporters [5]

ENOG410Y9WS ENOG410ZNX6, ENOG410ZJX3, ENOG410YNPF, ENOG410YAIR,

ENOG410YB7D, ENOG4110QZN, ENOG410YB3G, ENOG410XQAZ,

ENOG4111JQD, ENOG4111HM8, ENOG410ZPJF, ENOG410YDBU Anemia is a phenotype of colitis, increased iron uptake leads to anemic condition in host

Amino acid transporters [6]

 COG2610, COG0411, ENOG4111GBW Bacteria under an inflammatory state usually have a weak capacity to make nutrients on their own. To overcome the problem of nutrient deficiency, the bacteria instead transport them from the available environment through means such as tissue destruction or from the sites of inflammation.

Carbohydrate transporters ENOG410ZVEB ENOG410ZWA1, ENOG410XSHZ, ENOG410XNZG, ENOG41105XQ, ENOG410YA44 

ABC transporters COG0614, ENOG410ZUAG, ENOG410XPKQ, ENOG410XQ2F, ENOG410XQUM 

Calcium ion transport/

Hydrogen transport COG0387 

ammonium transporter COG0004 Inconclusive

formate transmembrane transporter COG2116 

Mucin degradation related genes significantly found to increase in KO mice group, may further reduce mucins in inflamed gut

Mucin degradation [7]

 COG4724, 

COG4193, 

COG0363 Degrade mucins from colonic epithelium

SusD family [7]

 ENOG410YEJG 

glycosyl hydrolase [8] 

 ENOG411272W involve in degradation of mucin O-linked glycans

Amino Acid Metabolism

Proline biosysthesis [9]

COG0014 Known to sustain intestinal stability and defend against IBD, genes are significantly increased in CO as compared to KO

Histidine biosynthesis [9]

COG0040, COG1387 histidine supplementation improved colitis condition in an IL-10-deficient cell transfer model of Crohn’s disease, similarly we found histidine biosynthesis genes increased in CO as compared to KO

Amidohydrolase (Histidine catabolic process) COG1228 In KO mice group, host gut is already inflamed, any further histidine degradation might aggravate colitis.

Arginine metabolic process/ Amoebiasis (05146)/catabolic activity COG0010 Amoebiasis is 

Gastroenteritis, leading to diarrhoea, caused by parasitic infection. Similar gene may aggravate colitis in KO mice group.

Information retrieved from KEGG pathway 05146

arginine dihydrolase [9] (arginine degradation) COG2235 In KO mice arginine degradation is enhanced. However, arginine supplementation has been demonstrated to maintain normal intestinal physiology and promote mucosal healing when inflammation affects the intestine.

Phyenylalanine, tyrosine and tryptophan biosysthesis [10]≠

COG0512 COG1465, COG4401 Phenylalanine has been reported to be persistently up-regulated and positively correlated with tyrosine in IBD patients even during induction of remission, especially in patients with CD

Phenylalanine metabolism [10]≠

 COG1473 

Tryptophan RNA-binding attenuator protein (Regulate tryptophan metabolism) ENOG411234W 

Lysine biosynthesis [11]

 COG0329, ENOG410ZK6H In pediatric IBD patients, it has been reported that lysine increased in fecal amino acid composition.

Carbohydrate Metabolism

Metabolize complex sugars COG1621, COG3345, COG2730, COG0296, COG3867 In CO group, the metabolic capacity of bacteria remains high, that’s why gut microbiome are able to metabolize both complex as well as simple sugar. Although in microenvironment of inflamed gut, resident microbiome may express genes that are capable of metabolizing simple sugars only as in KO mice group.

Metabolize simple sugars COG0191, COG0153, ENOG4111X3Y, ENOG410XV9T COG0149, COG0158, ENOG4111F99, ENOG410XQWR 

Cell Motility

Cell motility[12, 13]

 COG1724 In previous IBD studies on T-bet−/− Rag2−/−, resulting in colitis and TNBS-induced colitis mouse models, an increased capacity for bacterial pathogenesis has been observed. These models are found to exhibit a high abundance of genes related to cell motility.

Biofilm formation COG1734, ENOG4111QHP Essential for pathogenicity

arsR family transcriptional regulator [14]

 ENOG410XUWM Members of this family reported to involve in biofilm formation and virulence

protein phosphorylation (serine protein kinase) COG2766 Regulate several processes including cell-cell communication i.e. biofilm formation

Protein kinase ENOG410ZN6D 

Sulfur Metabolism

Sulfur metabolism, hydrogen sulfide biosynthetic process[15, 16]

 COG0155 hydrogen sulfide (H2S) is a phenotype of colitis, found to increase in KO mice group

Methane metabolism

Methane metabolism[17]

 COG1456 Increased methane production is reported in IBD patients, found to increase in KO mice group.

Restriction modification system is a robust system that provide defense to bacteria from viral DNA infection. KO mice group loose this defense system as compared to CO.

Restriction modification system related COGs [18] (prevent foreign (viral) DNA to invade or infect the bacterial cell) ENOG410ZVHQ, ENOG410Z7SX, COG1401, COG4268, ENOG410YBMI, ENOG4111V23, COG4823, ENOG410XV7B, ENOG410YTH4, ENOG410ZVTA ENOG411245M, ENOG410ZYXE, ENOG410YFNS Colitis phenotype often associated with increased abundance of bacteriophages.

Cell Wall Biosynthesis a

Alanine racemase (catalyzes L-alanine to D-alanine) COG3616 peptidoglycan biosynthesis

dTDP-4-dehydrorhamnose reductase COG1091 Participate in biosynthesis of the dTDP-L-rhamnose which is an important component of lipopolysaccharide (LPS)

Glycosyl transferase ENOG4111IEG, ENOG4111TT7, ENOG410ZVFD, ENOG4111SQ1, ENOG4111TNE, ENOG411290R, ENOG4111WRY, ENOG410ZWI8, ENOG410ZVET, ENOG410ZVJG ENOG410Y5QZ, ENOG410Y03U, ENOG4111FP0 involved in cell wall formation

Cell wall formation (Adds enolpyruvyl to UDP-N- acetylglucosamine) COG0766 Cell wall biosynthesis

Teichoic acid biosynthesis protein A (found within the cell wall of most Gram-positive bacteria) ENOG4112B0N Cell wall biosynthesis

Glycerolipid metabolism COG3345 (repeated) ENOG410XQJ1 Provide precursors for cell wall synthesis

outer membrane assembly protein (found in gram-negative bacteria) ENOG410XQ8P Cell wall biosynthesis

penicillin-binding protein 1C (peptidoglycan biosynthesis) COG4953 Cell wall biosynthesis

Nad-dependent epimerase dehydratase ( involve in lipopolysaccharide biosynthesis) ENOG4111J45 Cell wall biosynthesis

Cell wall resistance gene found to increase in KO mice group

Capsule polysaccharide COG3563 Important virulence factor for many pathogens

spore coat protein ENOG4111IEQ, ENOG4111KT8, ENOG41123YA confers resistance to peptidoglycan-breaking enzymes and noxious chemicals in KO mice group

Fatty Acid Metabolism a

Fatty acid biosynthesis COG0332 

Lipid biosysthesis (synthesize molecules that are typically precursors to membrane phospholipids) COG0615 

Nucleic Acid Metabolism a

Pyrimidine metabolism

(Tetrahydrofolate is a co-factor which is needed for the de-novo synthesis of thymidine monophosphate, required for the biosynthesis of bacterial DNA and RNA) COG1351 required for nucleic acid synthesis

Pyrimidine metabolism, Salvage pathway of pyrimidine, the catabolic breakdown and recycling of nucleoside compounds COG0295 required for nucleic acid synthesis

Phosphonoacetate hydrolase (phosphonoacetate + H2O acetate + phosphate) COG2824 Phosphate required for nucleic acid synthesis

DNA Replication & Repaira

DNA Replication COG0553 COG0328, COG1372, COG0358, COG0593, ENOG410ZJKI DNA Synthesis

RNA Biosynthesis Process ENOG410ZNQ9 RNA Synthesis

tRNA maturation COG0689 

Transposase ENOG410XT7Q, ENOG410ZXW1, ENOG4111GFI, COG2963, ENOG4111WA6, ENOG410ZUF1, COG1484, ENOG4111M0T, COG5659 

DNA Repair COG0350, COG0648 COG1197, COG0322,COG4294, ENOG410ZZVS, ENOG410YJ72 

Antibiotic Resistance found to increase in KO mice group.

#Establishment of competence for transformation (The process in which a naturally transformable bacterium acquires the ability to take up exogenous DNA. Competence allows for rapid adaptation and DNA repair of the cell) [19]

 COG4940 Horizontal gene transfer is a potential source of antibiotic resistance genes transfer

Conjugative transposition [19]

 ENOG410ZKPD 

RNA methylase (methylation, RNA function and antibiotic resistance) ENOG4111FKJ Antibiotic resistance

Regulator of multiple

antibiotic resistance ENOG4111TDN(repeat), ENOG4111IWY(repeat),COG4189 Antibiotic resistance

Multidrug Resistance protein COG2076 Antibiotic resistance

Transcriptional regulator, TetR family ENOG4111T38 ENOG4111NTM, ENOG41110YX Antibiotic resistance

Beta-lactamase[20]

 ENOG410ZWUI Antibiotic resistance, often known to aggravate colitis

PemK-like protein, Transcriptional modulator of MazE toxin, MazF[21]

 ENOG410ZMEC, ENOG4111SRX stabilize plasmids that carry drug-resistance determinants in important pathogens

Energy production related genes significantly increased in KO mice group b

Biotin metabolism (00780) COG1424 Coenzyme in carboxylation reactions

Nicotinate and nicotinamide metabolism (00760) 

 COG2816 Serve as coenzyme in oxidation-reduction reactions

electron transport and ATP synthesis ENOG410ZB0Z, ENOG4111NY9, COG1139, COG1773, ENOG41120JQ COG0043, COG0163, ENOG410XRT9, COG1304

ENOG4111I9P Energy Production

oxidation-reduction process COG1012, COG2055, ENOG410XS3N, COG1062 Energy Production

energy derivation by oxidation of organic compounds 

 COG1892 Energy Production

Citrate lyase is an enzyme which converts citrate to oxaloacetate.

(citrate fermentation� ultimately convert into D-lactate) COG3052 Energy Production

COGs/NOGs involved in regulatory processes, mainly found increased in CO mice group*

positive regulation of carbohydrate metabolic process COG2909 

Nucleotide-binding protein, involve in regulatory processes COG4271 

regulation of bacterial Mg2+ homeostasis COG0336 

Regulation of cell autolysis ENOG410XW9I 

ubiquitin mediated proteolysis (04120) COG0476 Degrade misfolded proteins

N-End Rule protein degradation COG2360, COG2127 COG1067 

Biological regulation (Serine threonine protein kinase) COG0515 

Intracellular signal transduction ENOG41101BA 

Sulfur relay system (04122) COG0607, ENOG410XPIQ Cell redox homeostatsis, Thiamin biosynthesis

Cobalamin biosynthesis COG1492 Interamolecular rearrangements

(for ex, methyl group transfer during methionine synthesis)

Regulation of RNA biosynthetic process ENOG410XRI9 

Unknown ENOG4111YI0, ENOG410XYYF ENOG410Y8FZ, ENOG410YX0Q ENOG4111XP0 

≠ Phenylalanine was persistently up-regulated and positively correlated with tyrosine in IBD patients even during induction of remission, especially in patients with CD. However, the pattern of increased aromatic amino acids (e.g. phenylalanine and tyrosine) and a decrease in branched chain amino acids (e.g. valine) is well known during catabolic conditions like sepsis and liver failure[10].

#The phenomenon of natural transformation has enabled bacterial populations to overcome great fluctuations in population dynamics and overcome the challenge of maintaining the population numbers during harsh and extreme environmental changes. During such conditions some bacterial genera spontaneously release DNA from the cells into the environment free to be taken up by the competent cells. The competent cells also respond to the changes in the environment and control the level of gene acquisition through natural transformation process.

a Our results have also demonstrated significant changes in various other pathways and gene abundances between the CO and KO groups. For example, the genes associated with “lipid metabolism” and “cell wall/membrane/envelope biogenesis” were found to be decreased while those related to “replication, recombination and repair” and “nucleotide transport and metabolism” were found to be increased in the KO as compared to CO group. This indicates that due to a reduced lipid metabolism the lipids were not available for cell wall/membrane/envelope biogenesis which led to bacterial endangered survival. Thus, bacteria might choose alternative means for handling this situation by increasing the replication rate to increase their numbers. In addition, due to an increased number of movable elements, i.e. transposase, the bacteria might allow mutational changes in their genome to adapt to an inflamed intestinal environment. Although the genes associated with “cell wall/membrane/envelope biogenesis” were found to be decreased in the KO group but the genes associated with beta-lactamase were found to be increased. This could provide a defense to the resident gut bacteria of the KO mice. The beta-lactamase produced by bacteria is known for its multi-resistance to β-lactam antibiotics. Recently it has been shown that increased β-lactamase production by gut bacteria might aggravate ulcerative colitis in patients [20]. Thus beta-lactamase may enhance colitis in Ap1m2-/- mice as well, but the mechanism remains unknown.

b Energy production related genes are significantly increased in KO mice group. In KO mice group, different transporters are found to increase indicating increased requirement of nutrient uptake. We hypothesize that more energy will be required for active nutrient transportation resulting into increased energy production in KO. 

*Mentioned genes are although involved in several regulatory processes, however, involvement of these genes remain inconclusive in colitis phenotype. 

In addition to including the above table in the revised manuscript, we have also made the following changes in the revised manuscript:

Page no. 7; line no. 153-155: Added “In addition to this, a threshold was set at the functional level such that only those significant functions were considered for the downstream analysis that have a mean abundance difference between CO & KO groups > 5.”

Page no. 14; line no. 277-282: Modified “Out of these 245 COGs, 156 were found to be increased while 89 were found to be decreased in the KO as compared to the CO group.” � “In total 245 functions at level 3 were found to be significantly altered (S4 Table). Out of these functions, 44 functions were found to have very low read counts (mean abundance difference between CO & KO groups < 5) and were not considered for further analysis. The details of the results obtained for the remaining 201 significantly altered functions are given in S5 Table. Out of these 201 COGs/NOGs, 128 were found to be increased while 73 were found to be decreased in the KO as compared to the CO group.”

Page no. 14; line no. 283-284: Added “A comprehensive literature survey was performed to see the effect of these functional alterations in IBD”

Page no. 14; line no. 299-301: Added “Additionally, genes related with amino acid biosynthesis viz, proline (COG0014) and histidine (COG0040, COG1387) were often found to be decreased in the KO as compared to the CO group.”

Page no. 15; Table 2: 

Genes involved in functions CO KO

Carbohydrate metabolism [13] f

Lipid metabolism f 

Cellwall/membrane/envelope biogenesis f 

Cell motility [12, 13] e

Mucin degradation [7] e

Replication, recombination and repair e 

Nucleotide transport and metabolism e 

Transposase e 

Amino acid transportation [6] e

Carbohydrate transportation e 

Iron transportation [5] e

Oxidative stress [2, 3] e

β-lactamase [20] e

Proline Biosynthesis [9]

Histidine Biosynthesis [9]

Arginine degradation [9]

1. Three functions added viz, proline biosysnthesis, histidine biosynthesis and arginine degradation. 

2. Two more columns added, one for CO group and another for KO group, to depict abundance of a particular function (Dark grey color indicates the abundance of a particular function is increasing and light grey color indicates the abundance of a particular function is decreasing). 

3. Added on the Table 2 legend “Dark grey color indicates the abundance of a particular function is increasing and light grey color indicates the abundance of a particular function is decreasing” (Page no. 15; line no. 308-309)

4. Footnote removed from Table 2 (Page no. 16; line no. 310-311)

Page no. 16; line no. 326-327: Added “Additionally, genes involved in degradation of arginine (COG0010, COG2235) and histidine (COG1228) were also found to be increased in the KO group as compared to the CO group.”

Page no. 17; line no. 333-334: Modified from “Out of the 245 significantly differing COGs functions, 98 mapped to five bacterial species” � “Out of the 201 significantly differing COGs/NOGs functions, 89 mapped to five bacterial species”

Page no. 21; line no. 410-420: Added “For the growth and maintenance of barrier function and mucosal integrity, amino acids are essentially required. Amino acid metabolism in microbial community of gut, plays significant role in physiology and nutrition of the host. We found genes related with proline and histidine biosynthesis decreased and arginine degradation genes increased in KO as compared to CO. In previous studies supplementation of histidine, proline and arginine has been reported to improve IBD [9]. Dietary proline supplementation improved the production of mucin, restored healthy microbiota, protected gut epithelium and encouraged mucosal rebuilding in DSS-treated rat [9]. Similarly, histidine supplementation improved colitis condition in an IL-10-deficient Crohn’s disease model. This improvement was done by regulation of NF-κB activation, followed by inhibition of the proinflammatory cytokine secretion by macrophages [9]. Additionally, changes in the metabolism of arginine have been reported in animal colitis models as well as IBD patients, and experimental colitis has been improved with arginine supplementation [9].” 

Page no. 22; line no. 460-464: Added “As is evident from the above discussion that a large number of the functional alterations between the CO and KO groups correlated well with the observed processes known to play a role in IBD. It is important to note that a few functional alternations still remained inconclusive. The analysis presented in the paper provides future directions for the design of more targeted experiments for the validation of the functional alterations due to the knockout of Ap1 gene leading to spontaneous colitis in mice.”

2. Further, some discussion on the etiology of the microbial changes would be welcome. 

We have made changes in manuscript on following lines:

Page no. 8; line no. 190-191: The increased abundance of Peptostreptococcaceae have been previously reported in IBD patients (line number in main manuscript, 181-182). 

Page no. 9; line no. 213-216: Class Deltaproteobacteria contains bacteria that reduce sulfate. The higher abundance of these bacteria is also known in IBD patients. Sulfate-reducing bacteria have been suggested to aggravate gastrointestinal diseases by making hydrogen sulfate (H2S) and other harmful by-products as well as by reducing beneficial metabolites, such as butyrate.

How specific are these for Ap1m2 KO mice, when compared to other mouse models of colitis? 

Table 2 presents the major microbial changes observed in different colitis mice models. It is evident from this table that the microbial changes observed in Ap1b-/- mice are more similar to those in DSS colitis mice model. With respect to bacteroides, in some studies its abundance is reported to be enhanced and in others it is reported as reduced [22]. However, in Ap1b-/- mice bacteroides have been found to be reduced. It is also evident from table 2 that in all previously reported mice models there has been an increased abundance of the members of class Gamaproteobacteria in colitis phenotype. However, in Ap1b-/- mice Gamaproteobacteria have been found to be decreased. Instead in Ap1b-/- mice we observed an increased abundance of the members of class Deltaproteobacteria. Just like the DSS mice model the Ap1b-/- mice model may serve as an extremely popular model for IBD studies.

Table 2: Microorganisms reported to associate with IBD in the mouse.

Type of disease or model Microorganisms Final effect Method used References

DSS colitis

(Chemically induced) Bacteroides distasonis, Clostridium ramosum, Akkermansia muciniphila, Enterobacteriaceae,

Desulfovibrio desulfuricans.

Erysipelotrichales 

 High abundance correlate with acute and chronic ulcerative colitis Culture method, T-RFLP, qPCR, metatranscriptomics, bacterial 16S rRNA gene amplicon sequencing

 [23-25]

Colitis in IL-10 deficient mice Enterobacteriaceae and adherent-invasive E. coli High abundance correlate with inflammation (Enterobacteriaceae) and cancer (E. coli) 16S rRNA gene amplicon sequencing

 [26, 27]

TNBS colitis

(Chemically induced) Enterobacteriaceae, Bacteroides High abundance correlate with inflammation RT-PCR, qPCR [28]

TRUC model Enterobacteriaceae,

Klebsiella pneumonia,

Proteus mirabilis High abundance correlate with inflammation 16S rRNA+

Whole metagenome shotgun sequencing [12]

Ap1b-/- mice Clostridiales (family, genera, species)

Desulfovibrionaceae (genera, species) 

Erysipelotrichaceae sp. High abundance correlate with spontaneous chronic colitis Whole metagenome shotgun sequencing Unpublished data (Our study)

3. Finally, the title of this manuscript ('Dysbiosis in Gut-Microbiome of Clathrin Adapter AP-1B Knockout Leading to Colitis in Mice), as well as the running title (Role of Gut-Microbiome in AP-1B Knockout Colitis), implies that dysbiosis leads to colitis in these mice, but there is no evidence presented to support such causality.

Main title:

“Association of Colitis with Gut-Microbiota Dysbiosis in Clathrin Adapter AP-1B Knockout Mice”

Running title:

“Dysbiosis of Gut-Microbiota in AP-1B Knockout mice.”

References

1. Takahashi D, Hase K, Kimura S, Nakatsu F, Ohmae M, Mandai Y, et al. The epithelia-specific membrane trafficking factor AP-1B controls gut immune homeostasis in mice. Gastroenterology. 2011;141(2):621-632. pmid: 21669204 

2. Körner H, Sofia HJ, Zumft WG. Phylogeny of the bacterial superfamily of Crp-Fnr transcription regulators: exploiting the metabolic spectrum by controlling alternative gene programs. FEMS Microbiol Rev. 2003;27(5):559-592. pmid: 14638413 

3. Sherrill C, Fahey RC. Import and Metabolism of Glutathione byStreptococcus mutans. J Bacteriol.1998;180(6):1454-1459. pmid: 9515913

4. Balmus I, Ciobica A, Trifan A, Stanciu C. The implications of oxidative stress and antioxidant therapies in Inflammatory Bowel Disease: Clinical aspects and animal models. Saudi J Gastoenterol. 2016;22(1):3-17. pmid: 26831601

5. Kaitha S, Bashir M, Ali T. Iron deficiency anemia in inflammatory bowel disease. World J Gastrointest Pathophysiol. 2015;6:62–72. pmid: 26301120

6. Morgan XC, Tickle TL, Sokol H, Gevers D, Devaney KL, Ward DV, et al. Dysfunction of the intestinal microbiome in inflammatory bowel disease and treatment. Genome Biol. 2012;13(9):R79. pmid: 23013615

7. Martens EC, Koropatkin NM, Smith TJ, Gordon JI. Complex glycan catabolism by the human gut microbiota: the Bacteroidetes Sus-like paradigm. J Biol Chem. 2009;284(37):24673-24677. pmid: 19553672

8. Tailford LE, Crost EH, Kavanaugh D, Juge N. Mucin glycan foraging in the human gut microbiome. Front Genet. 2015;6:81. pmid: 25852737

9. Liu Y, Wang X, Hu C-AA. Therapeutic potential of amino acids in inflammatory bowel disease. Nutrients. 2017;9(9):920. pmid: 28832517

10. Bjerrum JT, Steenholdt C, Ainsworth M, Nielsen OH, Reed MA, Atkins K, et al. Metabonomics uncovers a reversible proatherogenic lipid profile during infliximab therapy of inflammatory bowel disease. BMC medicine. 2017;15(1):184. pmid: 29032767

11. Bosch S, Struys EA, van Gaal N, Bakkali A, Jansen EW, Diederen K, et al. Fecal Amino Acid Analysis Can Discriminate De Novo Treatment-Naïve Pediatric Inflammatory Bowel Disease From Controls. J Pediatr Gastroenterol Nutr. 2018;66(5):773-778. pmid: 29112087

12. Rooks MG, Veiga P, Wardwell-Scott LH, Tickle T, Segata N, Michaud M, et al. Gut microbiome composition and function in experimental colitis during active disease and treatment-induced remission. ISME J. 2014;8(7):1403-1417. pmid: 24500617

13. Zheng H, Chen M, Li Y, Wang Y, Wei L, Liao Z, et al. Modulation of gut microbiome composition and function in experimental colitis treated with sulfasalazine. Front Microbiol. 2017;8:1703. pmid: 28936203

14. Wang Q, Chen M, Zhang W. A two-component signal transduction system contributes to the virulence of Riemerella anatipestifer. Journal of veterinary science. 2018;19(2):260-270. pmid: 29284206

15. Bhatia M. H2S and Inflammation: An Overview. In: Moore PK, Whiteman M, editors. Chemistry, Biochemistry and Pharmacology of Hydrogen Sulfide. Cham: Springer International Publishing. 2015; p. 165-180. pmid: 26162834

16. Szabó C. Hydrogen sulphide and its therapeutic potential. Nat Rev Drug Discov. 2007;6(11):917-935. pmid: 17948022

17. Monasta L, Pierobon C, Princivalle A, Martelossi S, Marcuzzi A, Pasini F, et al. Inflammatory bowel disease and patterns of volatile organic compounds in the exhaled breath of children: A case-control study using Ion Molecule Reaction-Mass Spectrometry. PloS one. 2017;12(8):e0184118. pmid: 28859138

18. Gogokhia L, Buhrke K, Bell R, Hoffman B, Brown DG, Hanke-Gogokhia C, et al. Expansion of bacteriophages is linked to aggravated intestinal inflammation and colitis. Cell Host Microbe. 2019;25(2):285-299. e8. pmid: 30763538

19. Huddleston JR. Horizontal gene transfer in the human gastrointestinal tract: potential spread of antibiotic resistance genes. Infect Drug Resist. 2014;7:167-176. pmid: 25018641

20. Skuja V, Derovs A, Pekarska K, Rudzite D, Lavrinovica E, Piekuse L, et al. Gut colonization with extended-spectrum β-lactamase-producing Enterobacteriaceae may increase disease activity in biologic-naive outpatients with ulcerative colitis: an interim analysis. Eur J Gastroenterol Hepatol. 2018;30(1):92-100. pmid: 29076938

21. Williams JJ, Hergenrother PJ. Artificial activation of toxin–antitoxin systems as an antibacterial strategy. Trends Microbiol. 2012;20(6):291-298. pmid: 22445361

22. Lupp C, Robertson ML, Wickham ME, Sekirov I, Champion OL, Gaynor EC, et al. Host-mediated inflammation disrupts the intestinal microbiota and promotes the overgrowth of Enterobacteriaceae. Cell Host Microbe. 2007;2(2):119-129. pmid: 18005726

23. Okayasu I, Hatakeyama S, Yamada M, Ohkusa T, Inagaki Y, Nakaya R. A novel method in the induction of reliable experimental acute and chronic ulcerative colitis in mice. Gastroenterology. 1990;98(3):694-702. pmid: 1688816

24. Håkansson Å, Tormo-Badia N, Baridi A, Xu J, Molin G, Hagslätt M-L, et al. Immunological alteration and changes of gut microbiota after dextran sulfate sodium (DSS) administration in mice. Clin Exp Med. 2015;15(1):107-120. pmid: 24414342

25. Schwab C, Berry D, Rauch I, Rennisch I, Ramesmayer J, Hainzl E, et al. Longitudinal study of murine microbiota activity and interactions with the host during acute inflammation and recovery. ISME J. 2014;8(5):1101-1114. pmid: 24401855

26. Arthur JC, Perez-Chanona E, Mühlbauer M, Tomkovich S, Uronis JM, Fan T-J, et al. Intestinal inflammation targets cancer-inducing activity of the microbiota. Science. 2012;338(6103):120-123. pmid: 22903521 

27. Yang I, Eibach D, Kops F, Brenneke B, Woltemate S, Schulze J, et al. Intestinal microbiota composition of interleukin-10 deficient C57BL/6J mice and susceptibility to Helicobacter hepaticus-induced colitis. PloS one. 2013;8(8):e70783. pmid: 23951007

28. Ettreiki C, Gadonna-Widehem P, Mangin I, Coëffier M, Delayre-Orthez C, Anton PM. Juvenile ferric iron prevents microbiota dysbiosis and colitis in adult rodents. World J Gastroenterol. 2012;18(21): 2619-2629. pmid: 22690070

---

## [Editor Report · Decision Letter 1]

7 Jan 2020

PONE-D-19-26965R1

Association of colitis with gut-microbiota dysbiosis in clathrin adapter AP-1B knockout mice

PLOS ONE

Dear Dr. Prakash,

Thank you for submitting your manuscript to PLOS ONE. After careful consideration, we feel that the manuscript has been greatly improved, and takes into account the comments made by the reviewers. However, we think that the manuscript still needs careful editing to correct for several mistakes. Just to take 2 examples, in the new manuscript (with modifications indicated in the text), the correct name for H2S is hydrogen sulfide, not hydrogen sulfate (line 215), and on line 340, the sentence "... hydrogen sulfide (H2S) is a phenotype of colitis" is unclear. Do you mean "Excessive hydrogen sulfide in the intestinal luminal content may participate to colitis" ? Please correct with the relevant literature.  Therefore, we invite you to submit a carefully edited revised version of the manuscript

We would appreciate receiving your revised manuscript by Feb 21 2020 11:59PM. To enhance the reproducibility of your results, we recommend that if applicable you deposit your laboratory protocols in protocols.io, where a protocol can be assigned its own identifier (DOI) such that it can be cited independently in the future. For instructions see: http://journals.plos.org/plosone/s/submission-guidelines#loc-laboratory-protocols

We look forward to receiving your revised manuscript.

Kind regards,

François Blachier, PhD

Academic Editor

PLOS ONE

---

## [Author Response · Author response to Decision Letter 1]

10 Jan 2020

Reviewer #1: We think that the manuscript still needs careful editing to correct for several mistakes. Just to take 2 examples, in the new manuscript (with modifications indicated in the text), the correct name for H2S is hydrogen sulfide, not hydrogen sulfate (line 215), and on line 340, the sentence "... hydrogen sulfide (H2S) is a phenotype of colitis" is unclear. Do you mean "Excessive hydrogen sulfide in the intestinal luminal content may participate to colitis" ? Please correct with the relevant literature. 

Reply: We thank you for giving us an opportunity to revise our manuscript. In the manuscript with track changes mode on, the following suggestions has been incorporated: 

Page no. 9; line no. 204-206: Modified “Sulfate-reducing bacteria have been suggested to aggravate gastrointestinal diseases by making hydrogen sulfate (H2S) and other harmful by-products as well as by reducing beneficial metabolites, such as butyrate” � “Sulfate-reducing bacteria have been suggested to aggravate gastrointestinal diseases by making hydrogen sulfide (H2S) and other harmful by-products as well as by reducing beneficial metabolites, such as butyrate” 

Page no. 16; line no. 311-312: Modified “It has been reported previously [29] that hydrogen sulfide (H2S) is a phenotype of colitis” “It has been reported previously that an excessive hydrogen sulfide (H2S) production is found in the feces of ulcerative colitis patients [29].” 

The reference for the above statement is already included in the manuscript. 

Page no. 18; line no. 342-344: Modified “In addition, H2S, a cytotoxic byproduct of SRB that exerts pro-inflammatory effects at high concentrations [45, 46], can exacerbate intestinal epithelial barrier damage and is a colitis phenotype.” “In addition, H2S, a cytotoxic byproduct of SRB that exerts pro-inflammatory effects at high concentrations [45, 46], can exacerbate intestinal epithelial barrier damage and its excessive production is reported in colitis condition [26].” 

In the above statement we have included reference [26] in the revised manuscript.

Apart from these changes, some other grammatical mistakes and typographical errors were also corrected as marked in the revised manuscript.

---

## [Editor Report · Decision Letter 2]

14 Jan 2020

Association of colitis with gut-microbiota dysbiosis in clathrin adapter AP-1B knockout mice

PONE-D-19-26965R2

Dear Dr. Prakash,

We are pleased to inform you that your manuscript has been judged scientifically suitable for publication and will be formally accepted for publication once it complies with all outstanding technical requirements.

With kind regards,

François Blachier, PhD

Academic Editor

PLOS ONE
---

## [Editor Report · Acceptance letter]

4 Mar 2020

PONE-D-19-26965R2 

Association of colitis with gut-microbiota dysbiosis in clathrin adapter AP-1B knockout mice. 

Dear Dr. Prakash:

I am pleased to inform you that your manuscript has been deemed suitable for publication in PLOS ONE. Congratulations! Your manuscript is now with our production department. 

With kind regards,

on behalf of

Dr. François Blachier 

Academic Editor

PLOS ONE